# High Temperature Mechanical Properties and Wear Performance of B$_4$C/Al7075 Metal Matrix Composites

**Sangmin Shin [1,2], Donghyun Lee [1,2], Yeong-Hwan Lee [1,2], Seongmin Ko [1,2], Hyeonjae Park [1], Sang-Bok Lee [1], Seungchan Cho [1]** 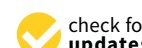 **, Yangdo Kim [2], Sang-Kwan Lee [1,\*] and Ilguk Jo [3,\*]**

[1] Composites Research Division, Korea Institute of Materials Science (KIMS), Changwon 51508, Korea; p996305s@kims.re.kr (S.S.); gsprada@kims.re.kr (D.L.); leeyh@kims.re.kr (Y.-H.L.); ksm0901@kims.re.kr (S.K.); guswodld@naver.com (H.P.); leesb@kims.re.kr (S.-B.L.); sccho@kims.re.kr (S.C.)
[2] Materials Science and Engineering, Pusan National University, Busan 46241, Korea; yangdo@pusan.ac.kr
[3] Division of Advanced Materials Engineering, Dong-Eui University, Busan 47340, Korea
\* Correspondence: lsk6167@kims.re.kr (S.-K.L.); ijo@deu.ac.kr (I.J.); Tel.: +82-51-890-1698 (I.J.)

**Abstract:** In this study, high volume fraction B$_4$C reinforced Al matrix composites were fabricated with a liquid pressing process. Microstructural analysis by scanning electron microscope and a transmission electron microscopy shows a uniform distribution of the B$_4$C reinforcement in the matrix, without any defects such as pore and unwanted reaction products. The compressive strength and wear properties of the Al7075 matrix and the composite were compared at room temperature, 100, 200, and 300 °C, respectively. The B$_4$C reinforced composite showed a very high ultimate compression strength (UCS) over 1.4 GPa at room temperature. The UCS gradually decreased as the temperature was increased, and the UCS of the composite at 300 °C was about one third of the UCS of the composite at room temperature. The fractography of the compressive test specimen revealed that the fracture mechanism of the composites was the brittle fracture mode at room temperature during the compression test. However, at the elevated temperature, AMCs had a mixed mode of a brittle and ductile fracture mechanism under the compressive load. The composite produced by a liquid pressing process also showed superior wear resistance compared with the Al matrix. The result of the wear test indicates that the wear loss of the Al matrix at 300 °C was two times higher than that of the AMCs, which is attributed to the formation of a mechanically mixed layer (MML) in the composites at the high temperature.

**Keywords:** Al matrix composites (AMCs); high volume fraction; liquid pressing process; fracture mechanism; mechanically mixed layer (MML)

## 1. Introduction

Aluminum based lightweight materials have a significant attraction in various industries, such as aircraft, automobile, and armor, for improving fuel efficiency and reducing CO$_2$ emissions through weight reduction of interior parts and body components [1–3]. However, it is hard to apply Al alloys directly in harsh working environment parts, such as engine blocks, powertrains, and braking systems, due to their relatively low strength and modulus and poor wear properties at elevated temperatures.

Ceramic particulate reinforced Al matrix composites (AMCs) provide a high strength to weight ratio, superior physical properties, thermal stability, good wear resistance, and other mechanical properties [4–9]. Moreover, the specific structural characteristics of AMCs affect the mechanical properties of composites [10]. Among the various ceramic materials, B$_4$C particulate is known as a promising reinforcement due to its high hardness, low density, and excellent thermal and chemical stability. A high-volume fraction (>40%) of B$_4$C reinforced AMC possesses a higher hardness and

compressive strength than that of AMCs with a lower volume fraction (<30%). However, producing high volume fraction AMCs is a hard challenge because of the poor wettability between the Al and the $B_4C$ particle [11].

AMCs can be produced by many techniques such, as stir casting, powder metallurgy, and infiltration. Generally, the stir casting method needs a second process to remove the defects and also has difficulties fabricating high volume fraction composites. For example, Soltani et al. [12], produced a 3 wt.% SiC reinforced Al composite by stir casting. The stir-cast specimen had defects such as air gaps, gas pores, and solidification shrinkage, which need to be removed. The powder metallurgy method can produce high volume fraction composites, but it needs a ceramic powder preform with a complicated process [13]. The liquid metal infiltration method would be excellent solution to fabricate high volume fraction AMCs with less pores and defects because the processing pressure facilitates the wetting between the liquid metal melt and ceramic reinforcement.

In the present study, a liquid pressing process was used to effectively fabricate high volume fraction $B_4C$ reinforced Al7075 composites [14,15]. The liquid pressing process is a new type of process, which is a promising way to fabricate AMCs based on low pressure that is near the theoretical minimum required pressing pressure at the melting temperature of the matrix metal. The microstructures of the Al matrix and composites were investigated to determine the feasibility of the liquid pressing process for producing high volume ceramic reinforced AMCs. This study also investigated the high temperature mechanical and wear properties of the matrix alloys and AMCs to establish the fracture and wear mechanism during the test at elevated temperatures.

## 2. Experimental Procedure

### 2.1. Materials and Methods

Al7075 alloy (2 mm plate, Kaiser aluminum, Foothill Ranch, CA, USA) as a matrix and $B_4C$ powder (5 and 40 μm, Dunhua Zhengxing Abrasive Co., Ltd., Dunhua, China) as reinforcements were used to fabricate AMCs through the liquid press processing. The mean particle size of the $B_4C$ reinforcement was measured using laser diffraction spectroscopy (Sympatec HELOS, Clausthal-Zellerfeld, Germany). The $B_4C$ particles with a 5 and 40 μm size were mixed together at a 1:1 weight ratio by a 3D Turbula mixer for one hour.

After mixing, an Al7075 plate and the mixed $B_4C$ particles were inserted into the steel mold, and then the temperature was elevated up to 800 °C under a low vacuum atmosphere. After the Al alloys melted, the mold was pressed in the mechanical press, and then, the pressing pressure was maintained until the temperature decreased under the solidus of Al7075. After the mold had cooled down completely, the composite was extracted and processed into specimens.

### 2.2. Characterization

The microstructure of the composites was evaluated using scanning electron microscopy (SEM, JSM-6610LV, JEOL, Tokyo, Japan) with 15 kV electron beam energy and 15 mm working distance. Field emission transmission electron microscopy (FE-TEM, JEM-2100F HR, JEOL, Tokyo, Japan) with 200 kV energy beam was also used to investigate the details of the composite microstructure. The average density of the composite was calculated using Archimedes principle from 5 measurements. The compressive properties of the composite were investigated using Gleeble3800 at room temperature, 100, 200, and 300 °C, respectively. Specimens with the dimensions of ϕ 10 mm × 12 mm were prepared for the compressive test, and the strain rate was $5 \times 10^{-4}$. In addition, in situ high temperature X-ray diffraction analysis was performed from room temperature up to 300 °C at a rate of 10 °C/min to observe the phase changes of the composites.

A wear test was carried out on a pin-on-disk type with steel pin counterpart (S45C) using the RB102-PD equipment (R&B Co. Ltd., Daejeon, Korea) at room temperature, 100, 200, and 300 °C, respectively. The samples were prepared into ϕ 30 mm × 10 mm size, and all the experiments were

carried out with a load of 40 N at a linear speed of 10 m/min for 200 m in atmospheric conditions. No lubricant was used in the test, and all samples were polished down to 0.25 μm. The friction coefficient between the disk and pin was measured from the frictional force with a sensor. After the wear test, the depth and width of the wear track were examined by a 2-dimensional profilometer (Dektak XT, Bruker, Gyeonggi-do, Korea). The worn surfaces and cross-sectional profiles were investigated using SEM and a field emission electron probe micro analyzer (FE-EPMA, JXA-8530F, JEOL, Tokyo, Japan) with a 15kV energy beam.

## 3. Results and Discussions

The average density of the $B_4C$ reinforced Al7075 composite specimen was 2.582 g/cm$^3$, which is almost similar to the theoretical density (2.588 g/cm$^3$) of the composite that was calculated by the rule of the mixture. The microstructure of the $B_4C$ reinforced Al7075 matrix composite is shown in Figure 1. The dark angular shape particles with a black color are the $B_4C$ reinforcement, and the grey colored area is the Al matrix. The result shows that the $B_4C$ particles were homogeneously dispersed without any defects or pores. In detail, 5 μm $B_4C$ particles were placed between the Al7075 matrix and the 40 μm $B_4C$ particles (yellow arrows). A higher infiltration pressure for the liquid pressing process is needed when the size of the reinforcement particle is smaller, which increases the manufacturing difficulty. However, aggregation or clusters of the 5 or 40 μm $B_4C$ particles were not observed. The detailed analysis on the interface between the matrix and the reinforcement by FE-TEM and transmission electron microscopy-energy dispersive spectrometry (TEM-EDS) are shown in Figure 1b,c. The interface was clean and free from any reaction product. From these results, it can be concluded that the composite with a high volume fraction $B_4C$ reinforcement was well fabricated through the liquid pressing process.

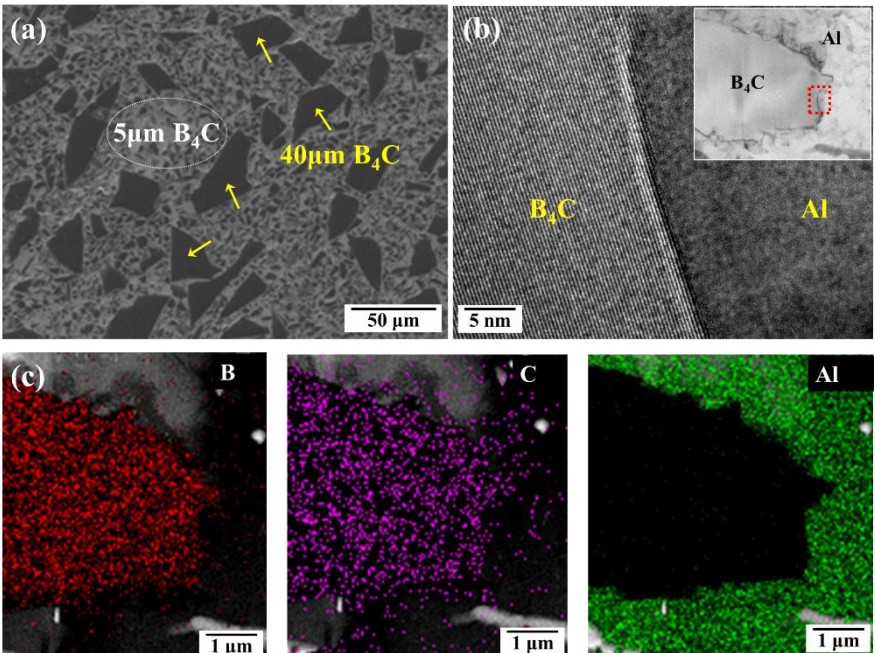

**Figure 1.** (**a**) Scanning electron microscopy (SEM), (**b**) field emission transmission electron microscopy (FE-TEM) and (**c**) transmission electron microscopy-energy dispersive spectrometry (TEM-EDS) images of bimodal $B_4C$/Al7075 Al matrix composites (AMCs) fabricated by the liquid pressing process.

Table 1 shows the mechanical properties of AMCs and Figure 2 shows the results of the compression tests from room temperature to elevated temperatures. As the temperature was increased, the ultimate compressive strength (UCS) of the composite decreased with the increase in the compressive fracture strain. Toughness of the composites also tended to decrease with increasing testing temperature,

although the toughness of the composites tested at 100 °C and 200 °C were of similar value. The UCS of the composite at the high temperatures was 936 MPa at 100 °C, 745 MPa at 200 °C, and 453 MPa at 300 °C, respectively. The compressive fracture strain at room temperature (8.3%) was gradually increased with the increment of the temperature, and it was about 11.8% at 300 °C. The stress-strain curve of the composite at room temperature shows an almost elastic behavior, and this indicates the brittle fracture of the specimen. The UCS of the composite at room temperature was 1444 MPa, and this value is higher than that of any $B_4C/Al$ composite in the literature [16–18]. Typically, a high-volume fraction of the reinforcement yields many defects, such as pores and clusters, resulting in a decrease in the compressive strength [19]. However, in this research, the compressive strength of a well-produced $B_4C/Al$ composite can exceed 1.4 GPa. The compressive yield strength (CYS) to UCS ratio (work hardening rate) decreased from 0.92 at room temperature to 0.77 at 300 °C. This result means that plastic deformation of the matrix at room temperature occurred less than at the high temperature [20]. This shows that the load transfer occurred efficiently, leading to the fracture of the ceramic reinforcement (brittle fracture mode) at room temperature. On the other hand, at the elevated temperature, the load transfer could not occur efficiently due to the softening of the matrix, and that was the reason for the reduction of the yield to UCS ratio. As a result, the occurrence of the brittle fracture and ductile fracture coexists at the high temperature. In addition, Figure 2b shows the yield to UCS ratio and the fracture strain changes for each temperature. As the temperature was increased, the yield to UCS ratio decreased, and the fracture strain increased.

**Table 1.** Mechanical properties of AMCs at variable temperature.

| Temperature (°C) | Compressive Yield Strength (MPa) | Ultimate Compressive Strength (MPa) | Strain (%) | Toughness (MPa%) |
|---|---|---|---|---|
| R.T. | 1328 | 1444 | 8.3 | 6516 |
| 100 | 796 | 936 | 8.3 | 4755 |
| 200 | 611 | 745 | 10.2 | 5218 |
| 300 | 349 | 453 | 11.8 | 3881 |

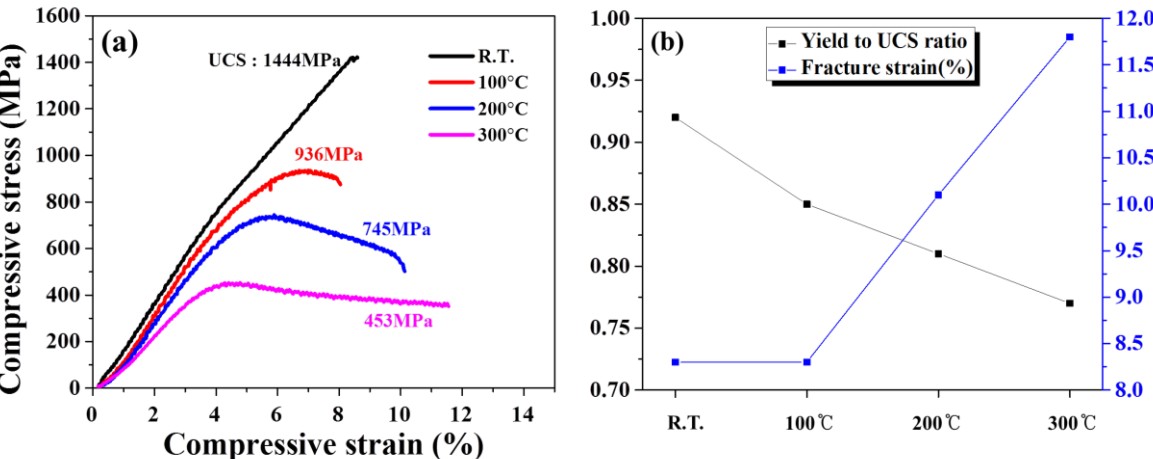

**Figure 2.** (**a**) Compressive properties, and (**b**) yield to UCS (ultimate compressive strength) ratio of the bimodal $B_4C/Al7075$ AMCs at room temperature and elevated temperatures.

Figure 3 shows the fracture surface of the compressive specimens with the different temperatures. At room temperature (Figure 3a), as indicated by yellow arrows, broken $B_4C$ particles with a cleavage fracture type were observed. Generally, a fracture occurs in high volume fraction reinforcement incorporated composites by de-bonding between the matrix and the reinforcement due to a poor bonding strength [21]. According to these, the fracture of the large $B_4C$ particles reflects the strong bonding strength in the composite and the well-bonded interface yielding an effective load transfer from the soft matrix to larger reinforcements. From the compressive test at 100 and 200 °C (Figure 3b,c),

a combination of cleavage fractures of B₄C particles and interfacial de-bonding between the matrix and the reinforcement was observed (combination fracture mode). Figure 3d shows that there were no significant brittle fracture phenomena at 300 °C. Instead, many tear ridges appeared to form in the matrix and the interface during the compressive fracture. Therefore, it could be concluded that the fracture mechanism changed from the brittle mode to the combination of the brittle and ductile mode because of the softening effect of the Al matrix at the elevated temperature of the composites [22].

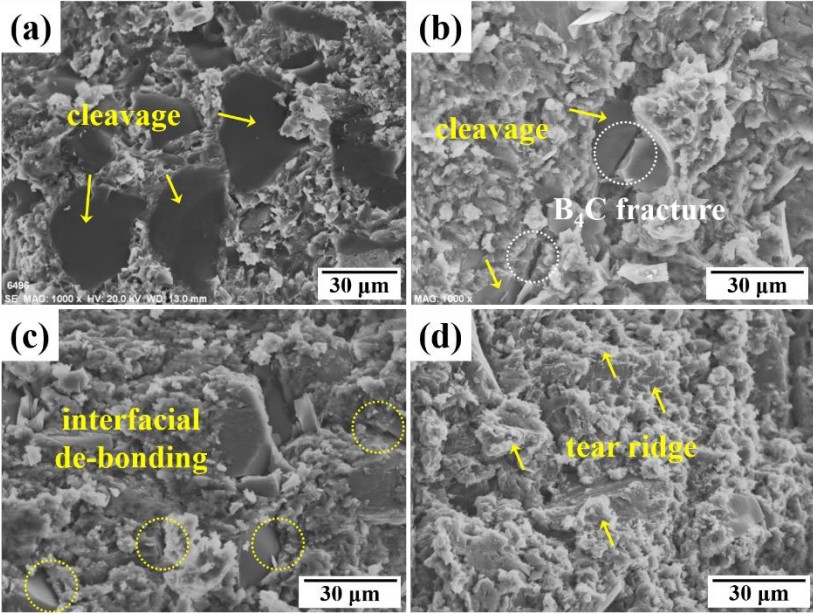

**Figure 3.** Fracture surface of the composite after the compressive test at (**a**) room temperature, (**b**) 100 °C, (**c**) 200 °C, and (**d**) 300 °C.

The crystalline phase patterns of the composites obtained by high temperature x-ray diffractometer (HT-XRD) are shown in Figure 4. As can be seen from the results, the formation of the new phase did not occur by the increase in temperature. However, with the increase in the temperature, the Al phase peak gradually shifted to the lower angle (lattice thermal expansion), which was considered to be the softening of the Al matrix by the high temperature. Topin et al. [23] showed that the matrix in the composite had a bulk effect with the load transfer and also had a surface effect that suppressed the particle movements. When the temperature is low, the matrix is strong, and the particles translocate a high load under a quasi-static load. At an elevated temperature, the matrix softens, and the particles can move easily under the quasi-static load, leading to a decrease in the UCS and an increase in the fracture strain [24].

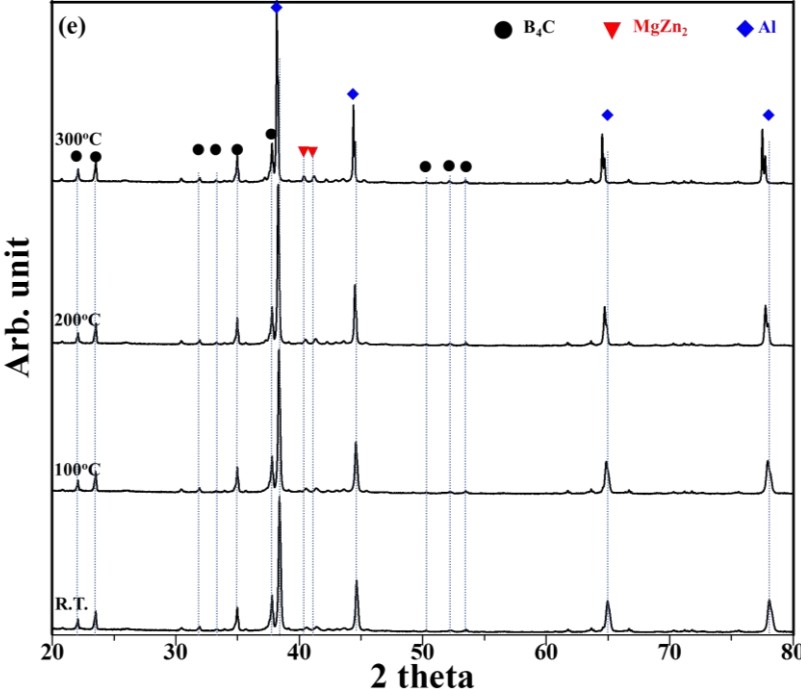

**Figure 4.** In situ high temperature-x-ray diffractometer (HT-XRD) results of the AMCs at elevated temperature.

Figure 5a,b shows the coefficient of friction (COF) of the Al7075 matrix and the composite, respectively, from the wear test at 25 °C and the elevated temperatures. The COF values of the Al7075 and the AMCs were in the range of 0.4–0.6 μ. The COF of both materials tended to increase as the temperature was increased, and this is attributed to the rough contact surface during the wear test. The wear depth and width measured by a 2-dimensional profilometer are shown in Figure 5c,d. The wear depth and width of the Al7075 specimen at room temperature (215.6 μm; 1.81 mm) were much larger than that of the composite specimen (17.1 μm; 1.60 mm), respectively. At the elevated temperature, the wear depth and width of the Al7075 at 300 °C (301 μm; 1.73 mm) were also larger than those of the composite (120.2 μm; 2.2 mm). In addition, the wear loss of Al7075 (26.1 mg) was 10 times higher than that of the AMCs (2 mg) at room temperature and about 2.5 times higher than that of the AMCs at 300 °C (Al7075: 53.5 mg; AMCs: 24.2 mg). These results indicate that the AMCs have a superior abrasion resistance compared with Al7075 matrix at each temperature, and the AMCs have a higher thermal stability due to the uniformly distributed $B_4C$ particles. The average wear depth of the composites at room temperature was 9.8 μm; however, the value of the average wear depth was increased (47.3 μm) at 100 °C. At 100 °C, due to the large particle debris on the surface, the abrasion of the composite was accelerated. Generally, as the temperature was increased, the wear depth increased exponentially; however, the wear depth of the AMCs increased logarithmically at the higher temperatures (200 °C: 64.3 μm; 300 °C: 64.4 μm).

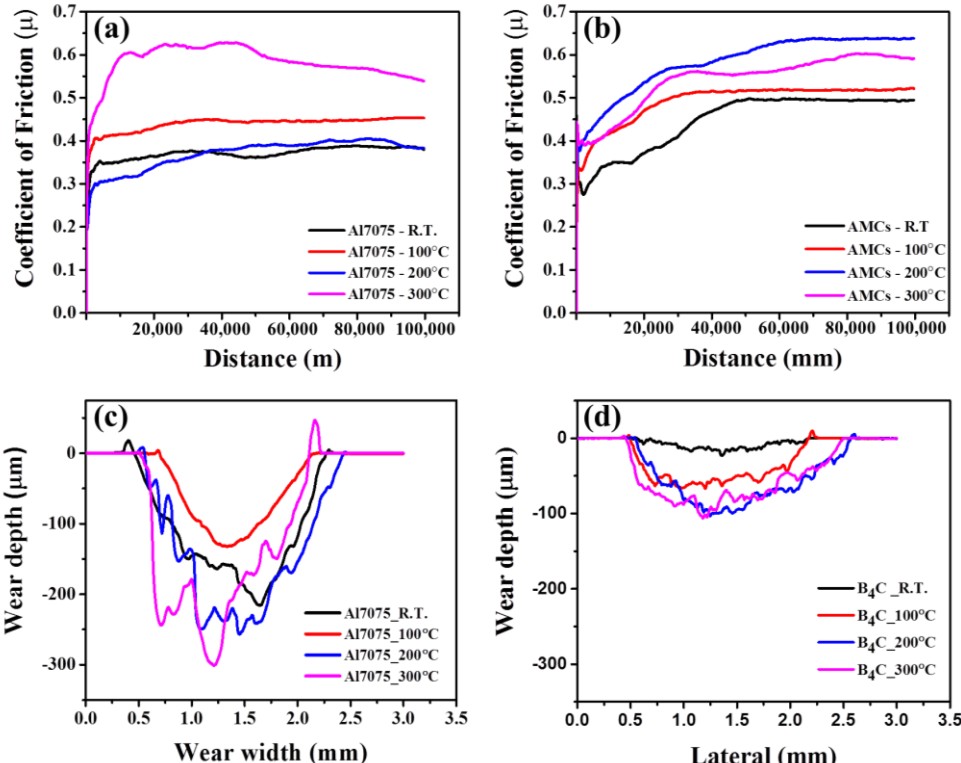

**Figure 5.** COF after the wear test (**a**) Al7075, (**b**) AMCs, depth and width of wear test, (**c**) Al7075, and (**d**) AMCs.

The microstructure of the worn surface of the AMCs and counter materials was investigated using SEM. Low magnification SEM images of the counterpart worn surface are shown in Figure 6a,b. Parallel sliding marks can be observed from the worn surface of the pin tip along the sliding direction, and the surface is flat and smooth with small grooves at room temperature. In the harsh test conditions (40 N at 300 °C), the abrasion of the $B_4C$ particles and the delamination layers of the steel pin were observed. As the counter material started to contact harshly with the surface of the composite specimen, it started to make large grooves on the surface of the pin as the sliding time increased. Figure 6c–f shows the SEM images of the worn surface of the AMCs. As can be seen from Figure 6c,d, the stress concentration mainly occurred around the large $B_4C$ reinforcement, and fractures of the large $B_4C$ particles were observed. Over 200 °C, the matrix became soft, and the bonding force between the matrix and reinforcement decreased, causing other wear phenomena compared with the lower temperature test. Therefore, dislodged particles and de-bonded (Figure 6e,f) areas came out, and those could cause the large debris and delamination parts on the worn surface. The large debris and delamination area caused the rough surface, which accelerated the wear [25].

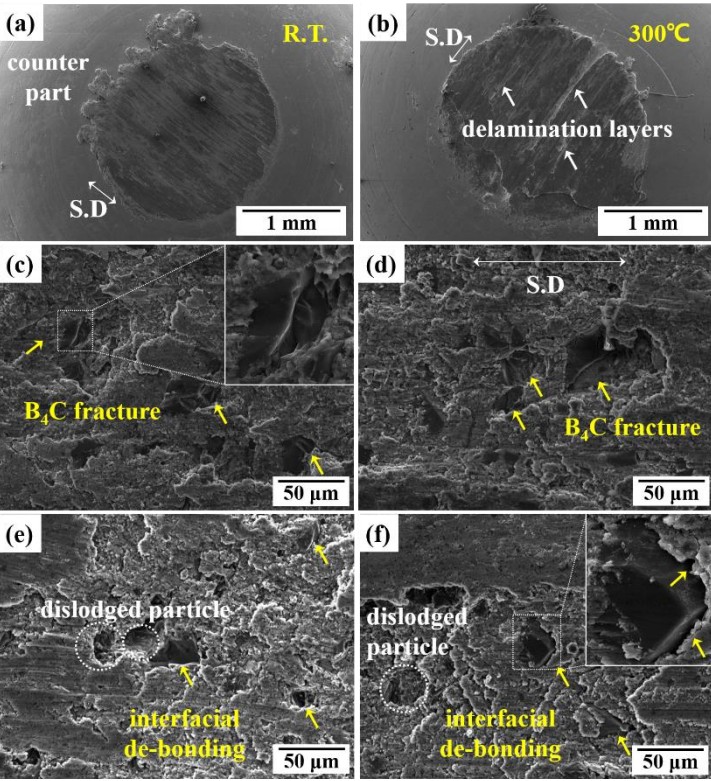

**Figure 6.** SEM images of the counter materials (steel pin) at (**a**) room temperature, (**b**) 300 °C, and worn surface of the composite after wear test at (**c**)room temperature, (**d**) 100 °C, (**e**) 200 °C, and (**f**) 300 °C.

Figure 7 shows the cross-sectional EPMA analysis of the wear test specimen from room temperature to 300 °C. As confirmed from Figure 7, the wear of the composite with the breakage of the 40 μm $B_4C$ particles occurred at room temperature and 100 °C due to the stress concentration. However, the fractured particles rarely detached, and they were physically well bonded with the matrix which is in accordance with the compressive test results. As the temperature was increased over 200 °C (Figure 7c,d), cracks in the matrix occurred, and broken particles were released from the matrix. As the wear progressed, the debris became finer, and the counterpart material and matrix were abraded together, and they reacted with oxygen to form a new layer on the surface. It is called a mechanically mixed layer (MML), and the MML commonly forms on the worn surface of metal matrix composites [26–29]. The MML is an oxide layer that is brittle and hard; thus, the MML could improve the wear resistance. However, the MML was not observed in Figure 7a,b. This is because the large $B_4C$ particles were placed on the worn surface of the AMCs; thus, the surface could not form the oxide layer. However, large $B_4C$ particles were finely crushed, and the particles were mixed with the matrix and the counterpart materials at elevated temperatures. Therefore, Al7075 and iron that were oxidized were distributed on the surface, and the MML started to form reacting with the oxygen. The MML formed with a thickness of about 5 μm at 200 °C, and the thickness of the MML was approximately 13 μm at 300 °C. From Figure 7, it can be seen that the wear depth of the composite did not increase exponentially at the higher temperatures. Thus, we can have deduced that the formation of the MML at a temperature over 200 °C could improve the wear resistance of the composite [30,31].

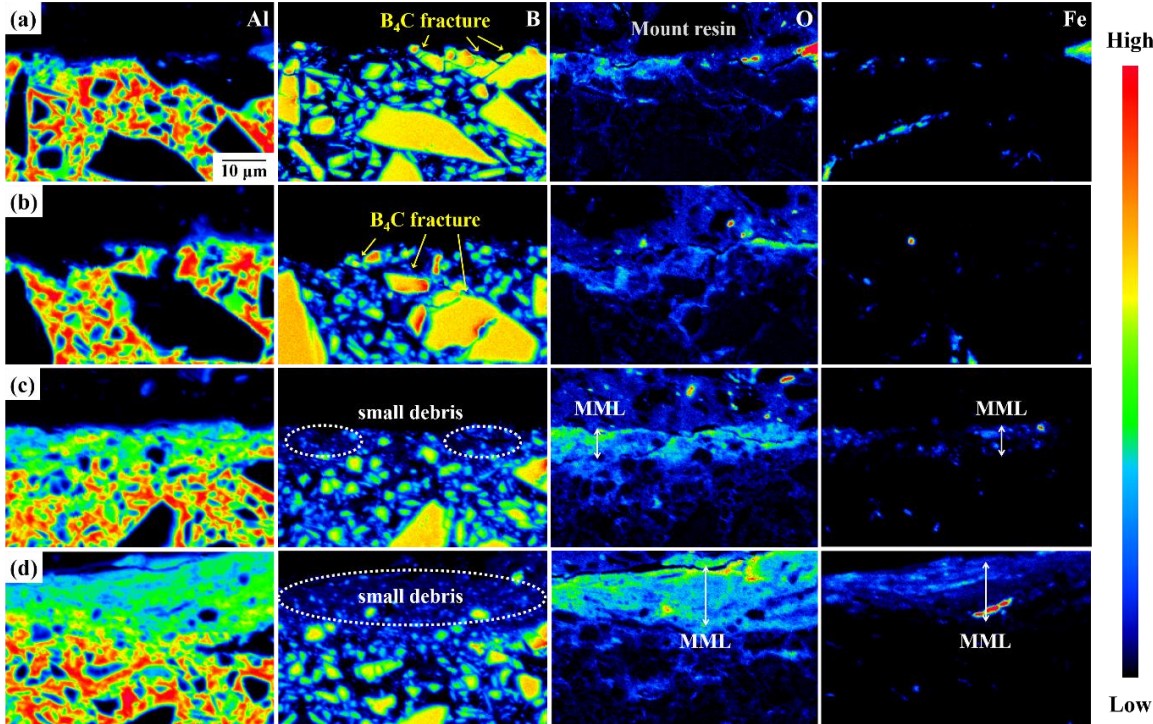

**Figure 7.** Cross sectional electron probe micro analyzer (EPMA) analysis images after the wear test(same magnification): (**a**) room temperature, (**b**) 100 °C, (**c**) 200 °C, and (**d**) 300 °C.

## 4. Conclusions

This study investigated the mechanical properties of high volume fraction B$_4$C/Al7075 AMCs fabricated by a liquid pressing process, and the results are as follows.

(1) High volume fraction B$_4$C reinforced Al7075 composites were successfully fabricated with a liquid pressing process. The real density of the specimen was almost similar to the theoretical density because the microstructural analysis revealed that no pores existed, and the interface between the matrix and the reinforcement was clean.

(2) From the high temperature compressive test, the highest UCS of the composite was 1.44 GPa at room temperature, and the compressive strength was decreased at elevated temperatures because of the softening of the matrix. The composite had a brittle fracture mode at room temperature and a mixed mode of a brittle and ductile fracture mechanism at the elevated temperatures under the compressive load.

(3) From the wear test, it was confirmed that the abrasion was accelerated due to the detachment of the particles or the delaminated layers. The average depth of the wear test at room temperature was 9.8 μm and 47.3 μm at 100 °C, which was an increase of about five times. However, at 200 °C (64.3 μm) and 300 °C (64.4 μm), the average depth was increased only 1.5 times compared to the result at 100 °C. Formation of a mechanically mixed layer at a temperature over 200 °C could improve the wear resistance of the composite at the elevated temperature.

**Author Contributions:** Conceptualization, I.J.; Investigation, D.L., Y.-H.L., S.K. and H.P.; Methodology, S.-B.L. and S.C.; Project administration, S.-K.L.; Resources, Y.K.; Supervision, S.-K.L. and I.J.; Visualization, I.J.; Writing—original draft, S.S.; Writing—review & editing, I.J.

**Funding:** This research was funded by the Korea Institute of Materials Science (KIMS) Fundamental Research Program (PNK6160) and by the Ministry of Science, ICT, and Future Planning, Korea, the National Research Foundation of Korea (NRF) grant (No. 2014M3C1A9060722). The authors are grateful for these grants.

**Conflicts of Interest:** The authors declare no conflict of interest.

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
