# Peer review of "High Temperature Mechanical Properties and Wear Performance of B4C/Al7075 Metal Matrix Composites"

_metals, doi:10.3390/met9101108_

Round 1
Reviewer 1 Report
The paper is focused in the mechanical properties and wear performance of B4C/Al7075 Metal Matrix Composites. The topic falls within the scope of the journal. The data are properly presented and discussed. I recommend its publication after the following revisions:
Experimental details for SEM/TEM analyses (such as working distance and energy of beam) should be reported. I recommend to determine the stored energy up to the fracture from the integration of the stress vs strain All the mechanical properties at variable temperature could be presented in an additional table. In the Introduction, the authors should better highlight the influence of the mechanical properties with the specific structural characteristics of composite materials as evidenced in recent reviews [Journal of Materials Research and Technology 8, 2019, 3347-3356; Current Opinion in Colloid & Interface Science, 35, 2018, 42]
Author Response
Thank you for the detailed suggestions and comments to our manuscript. These reviews have been very helpful in enhancing our paper. Below is a detailed discussion of the changes we have made to address the reviewer comments.
Detailed Comments for Author:
Reviewer 1
Comment 1. Experimental details for SEM/TEM analyses (such as working distance and energy of beam) should be reported.
→ The electron beam energy of the SEM and the EPMA were 15kV with the 15mm working distance. TEM analysis was carried out under the beam energy of 200kV. As commented, we supplemented the contents including detailed information of SEM, EPMA and TEM equipment.
Comment 2. I recommend to determine the stored energy up to the fracture from the integration of the stress vs strain All the mechanical properties at variable temperature could be presented in an additional table.
→ We added the table with summary of mechanical properties at variable temperature including calculated stored energy (toughness) using our law data. The stored energy tended to decrease when the temperature increased.
Comment 3. In the Introduction, the authors should better highlight the influence of the mechanical properties with the specific structural characteristics of composite materials as evidenced in recent reviews [Journal of Materials Research and Technology 8, 2019, 3347-3356; Current Opinion in Colloid & Interface Science, 35, 2018, 42]
→ Thank you for your comment and suggested recent paper (Characterization of Al-7075 metal matrix composites: a review, Journal of Materials Research and Technology 8, 2019, 3347-3356) is very helpful in our paper, so we cited the article in the introduction section (reference #10).

Reviewer 2 Report
Authors discuss the results of their works on Al matrix composites enriched with the high volume fraction of B4C, which were produced with a liquid pressing process. The microstructure changes of composites and the results of tests compressive strength and wear properties of the Al7075 matrix and the composite were compared at room temperature, 373, 473, and 573 K, respectively. Results of these works revealed that the wear resistance of the composite produced by a liquid pressing process was superior than the Al matrix.
The paper requires the following amendments:
General remark: SI units should be used in this article.
Abstract:
Short information about the aims of carried out investigations should be added.
Results and Disscusion:
TEM images: what indicates that this is B4C?
How do the hardness and Young of studied samples change?
Figure 4: the readability of both XRD patterns is low.
Did the authors carry out XPS studies?
In my opinion, the reviewed paper requires the minor alterations before being recommended for the publication as the article in the Metals.
Author Response
Thank you for compliments and comments to the manuscript. Your sincere reviews have been very helpful to our research. Below is our reply to your comments.
Detailed Comments for Author:
Reviewer 2
Comment 1. TEM images: what indicates that this is B4C?
→ The authors carefully revised the Figure 1 accordingly to avoid any confusion. Inset of Figure 1b indicates the low magnification TEM image of high resolution TEM image (Figure 1b). Highly aligned lattice fringe indicates the ceramic structure of B4C. Energy dispersive spectroscopy (EDS) element mapping images (Al, B, C) of the B4C/Al7075 composite shows present of B4C reinforcement in Al7075 matrix after liquid pressing process (Figure 1c).
Comment 2. How do the hardness and Young of studied samples change?
→ We studied about hardness at room temperature only. We are going to study hardness of B4C/Al7075 at elevated temperature in the future study to correlate the compressive test results. Young’s moduli of the B4C/Al7075 composites were deceased with increasing testing temperature as shown in Figure 2a, which confirmed decreased slope of stress-strain curve.
Comment 3. Figure 4: the readability of both XRD patterns is low.
→ Thanks for the delicate comment. We revise the XRD graph and increased readability.
Comment 4. Did the authors carry out XPS studies?
→ We didn’t carry out XPS studies in the current paper. However, it would be very helpful for understanding interface and quantitative research by using XPS as commented by reviewer. We hope to investigate detailed wear mechanism using XPS for the future work and submit other manuscript in the near future.
